# OFFLINE VISUAL REPRESENTATION LEARNING FOR EMBODIED NAVIGATION

**Karmesh Yadav**[1] **Ram Ramrakhya**[2] **Arjun Majumdar**[2] **Vincent-Pierre Berges**[1]
**Sachit Kuhar**[2] **Dhruv Batra**[1,2] **Alexei Baevski**[1] **Oleksandr Maksymets**[1]
[1]Meta AI Research [2]Georgia Institute of Technology
{karmeshyadav,vincentpierre,abaevski,maksymets}@meta.com
{ram.ramrakhya,arjun.majumdar,skuhar6,dbatra}@gatech.edu

## ABSTRACT

How should we learn visual representations for embodied agents that must see *and* move? The status quo is *tabula rasa in vivo*, *i.e.* learning visual representations from scratch while also learning to move, potentially augmented with auxiliary tasks (*e.g.* predicting the action taken between two successive observations). In this paper, we show that an alternative 2-stage strategy is *far* more effective: (1) offline pretraining of visual representations with self-supervised learning (SSL) using large-scale pre-rendered **images of indoor environments** (Omnidata (Eftekhar et al., 2021)), and (2) online finetuning of visuomotor representations on specific tasks with image augmentations under **long learning schedules**. We call this method Offline Visual Representation Learning (OVRL). We conduct large-scale experiments – on 3 different 3D datasets (Gibson, HM3D, MP3D), 2 tasks (IMAGENAV, OBJECTNAV), and 2 policy learning algorithms (RL, IL) – and find that the OVRL representations lead to significant across-the-board improvements in state of art, on IMAGENAV from 29.2% to 54.2% (+25% absolute, 86% relative) and on OBJECTNAV from 18.1% to 23.2% (+5.1% absolute, 28% relative). Importantly, both results were achieved by the same visual encoder generalizing to datasets that were not seen during pretraining. While the benefits of pretraining sometimes diminish (or entirely disappear) with long finetuning schedules, we find that OVRL's performance gains continue to *increase (not decrease) as the agent is trained for 2 billion frames of experience*.

## 1 INTRODUCTION

We are interested in teaching embodied AI agents to see (*i.e.* understand the structure and semantics of their environments) and move (*i.e.* strategically explore and navigate to accomplish goals). This endeavour is of fundamental importance from a practical perspective (*e.g.* for building home assistant robots) and a scientific perspective (*e.g.* what are the right visuomotor inductive biases?).

Broadly speaking, three different goal specifications have emerged in the embodied visual navigation literature: 1) point-goal navigation (POINTNAV (Anderson et al., 2018)), where an agent must navigate to a relative goal coordinate ('*go to* $\Delta x$, $\Delta y$'), 2) image-goal navigation (IMAGENAV (Chaplot et al., 2020b)), where an agent must navigate to the location of a goal image ('*find where **this** image was taken*') and 3) object-goal navigation (OBJECTNAV (Batra et al., 2020)), where an agent must navigate to an instance of a goal category ('*find bed*') – all in a new environment without a prebuilt map.

The status quo for solving these visual navigation tasks is to train agents *tabula rasa*, *i.e.* visual representations and navigation policies are trained from scratch for each specific task, optionally augmented with auxiliary tasks (e.g., predicting the action taken between two successive observations). This line of work has yielded some very exciting results. For instance, Wijmans et al. (2020) showed that the POINTNAV (Anderson et al., 2018) task situated in Gibson (Xia et al., 2018) environments can reach 99.6% success rate with reinforcement learning (RL) by training an agent for 2.5 billion steps in the Habitat simulator (Savva et al., 2019); Ramakrishnan et al. (2021b) further sharpened this result and achieved 100% success on this dataset. Similarly, Jaderberg et al. (2017) achieved near-human-performance in certain maze exploration tasks in DM Control Tassa et al. (2018) by augmenting deep RL with auxillary tasks.

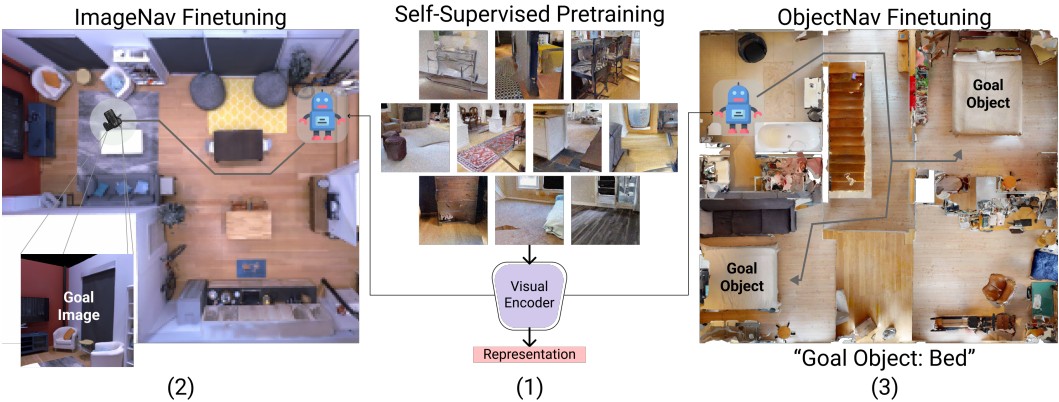

Figure 1: OVRL consists of two steps: (1) offline pretraining of the visual representations using large-scale pre-rendered images of indoor environments using DINO (Caron et al., 2021), and (2, 3) downstream finetuning of the visuomotor representations on the IMAGENAV and OBJECTNAV task.

However, for tasks requiring semantic scene understanding, results have been much weaker. The winning entries of the 2020 and 2021 OBJECTNAV challenges attained 18% (Chaplot et al., 2020a) and 24% (Ye et al., 2020) success rate respectively. In IMAGENAV, the best known result (in the single-RGB-camera setting) is 29.2% success rate (Al-Halah et al., 2022).

Clearly, what's true in 2D image understanding (Girshick et al., 2014) is also true in embodied scene understanding – *representations matter*. So how should we learn useful visual representations for embodied tasks?

In this paper, we propose Offline Visual Representation Learning (OVRL). As the name suggests, OVRL (Fig 1) divides visuomotor learning into two stages (Stooke et al., 2021): 1) pretraining visual representation offline and 2) downstream finetuning. The offline representation learning stage involves using a self-supervised learning (SSL) technique (DINO (Caron et al., 2021)) to train a vision model on a large-scale pre-rendered dataset of images from indoor environments called Omnidata (Eftekhar et al., 2021). In downstream finetuning, these representations are finetuned on individual task like IMAGENAV and OBJECTNAV in the Habitat simulator (Savva et al., 2019).

We find that OVRL significantly outperforms the tabula rasa status quo. Specifically, on IMAGE-NAV we advance the state of art from 29.2% (Al-Halah et al., 2022) to 54.2% (+25% absolute, 86% relative) in the single RGB camera setting on the Gibson dataset (Xia et al., 2018). On OBJECT-NAV, we advance state of art from 18.1% (Khandelwal et al., 2022) to 23.2% (+5.1% absolute, 28% relative) in the RGBD camera + known-pose setting. Importantly, we observe that the same pretrained vision model generalizes across different scene datasets. Specifically, in OBJECTNAV, OVRL outperforms our imitation learning (IL) from-scratch baseline (Ramrakhya et al., 2022) by 5.3%, which is impressive because the OVRL encoder *never saw MP3D scenes (Chang et al., 2017) during pretraining*. Performance gains of pretrained models are known to diminish (or entirely disappear) when finetuned with long schedules (He et al., 2019). Surprisingly, we find that the benefits of OVRL pretraining are not only sustained, but continue to *increase* (rather than decrease) over 2 billion frames of training on IMAGENAV using the HM3D (Ramakrishnan et al., 2021b) dataset; this suggests a significant rethinking might be needed in what we consider to be 'standard' training schedules for these tasks. Finally, we conduct extensive empirical ablations of OVRL's different components and find that finetuning the encoder with image augmentations is quite important for achieving good performance.

## 2  RELATED WORK

**Self Supervised Learning (SSL) in RL**: Prior work (Laskin et al., 2020a) has proposed contrastive learning (with image augmentations) as an auxiliary loss with RL, although it was later shown that the performance boost was due to image augmentation (Laskin et al., 2020b). CPC (Oord et al., 2018), CPC|Action (Guo et al., 2018) and ST-Dim (Anand et al., 2019) propose different variants of temporal contrastive losses, however these methods add complexity and require a sequence of images for training. ATC (Stooke et al., 2021) demonstrated the first decoupling of the representation learning and RL objective, using only pairs of images to train a temporal contrast objective. In comparison, our method does not require any form of temporal objective and is capable of learn-

ing representations from an IID collection of images. PBL (Guo et al., 2020), SPR (Schwarzer et al., 2020) and SGI (Schwarzer et al., 2021) employ non-contrastive temporal losses similar to BYOL (Grill et al., 2020), however they require extra loss terms to prevent their representation from collapsing, which our method doesn't need.

**Visual Navigation**: Both classical SLAM-based (Chaplot et al., 2020b; Hahn et al., 2021) as well as learning-based (Maksymets et al., 2021b; Mezghani et al., 2021) approaches have been proposed for embodied visual navigation. End-to-end learning methods typically use fewer hand-crafted modules and have shown more promise. Memory-Augmented RL (Mezghani et al., 2021) uses an attention-based model that leverages episodic memory to learn navigation and obtains SOTA results in IMA-GENAV with 4 RGB cameras. In comparison, we use a simpler model architecture while achieving higher performance. On the single camera setup, (Al-Halah et al., 2022) improves performance using a combination of a goal-view reward and goal-view sampling. We find that using this reward and view sampling leads to further improvements for OVRL models as well.

Similarly, end-to-end RL methods exist for OBJECTNAV and use data augmentation (Maksymets et al., 2021a) and auxiliary rewards (Ye et al., 2021) to improve generalization. In contrast, modular methods (Chaplot et al., 2020a), (Ramakrishnan et al., 2022) disentangle navigation and semantic mapping. Recently, a competitive imitation learning approach (Ramrakhya et al., 2022) powered by large scale dataset was proposed, which we build upon. (Mousavian et al., 2019) improves visual representations by including semantic segmentations, while we focus on RGB representations.

**SSL in Embodied AI**: EmbCLIP (Khandelwal et al., 2022) showed that using the CLIP encoder can provide useful representations for EAI tasks. CLIP is pretrained on an unreleased dataset of 400M image-caption pairs (WebImageText dataset (WIT)). In contrast, we pretrain on a much smaller (14.5M images) and public dataset called Omnidata Starter Dataset (Eftekhar et al., 2021). CRL (Du et al., 2021) proposes learning visual representations online using samples collected with a curiosity-based navigation policy that is incentivized to find images with a high SSL loss; we compare against a scaled-up version of CRL in our experiments and find that OVRL significantly outperforms it. EPC (Ramakrishnan et al., 2021a) used self-supervised learning to learn environment level representations by training to predict the missing zones in a zone segmented video sequence; however this requires pose information, OVRL pre-training does not (making it potentialy applicable to videos from the web in the future). Lastly, works from Ye et al. (2020; 2021) have showed that using auxiliary objectives during training can help on tasks like POINTNAV and OBJECTNAV. OVRL outperforms these results on OBJECTNAV without using any auxiliary losses, leaving open possible future improvements by combining the two ideas.

## 3 APPROACH

In this section we describe OVRL, our two-stage learning approach, which consists of an encoder pretraining step using DINO, followed by downstream policy learning in Habitat Simulator on the IMAGENAV and OBJECTNAV tasks.

### 3.1 SELF-SUPERVISED PRETRAINING

We pretrain our visual encoder using DINO (Caron et al., 2021), a recently proposed self-supervised learning (SSL) algorithm. DINO uses knowledge distillation, where a student network is trained to match the outputs of a teacher network. As illustrated in Fig 2 (left), an input image $x$ is first transformed using data augmentations. Specifically, we use the multi-crop data augmentation strategy introduced in Caron et al. (2020) to produce two global views ($x_1^g$ and $x_2^g$) at a $224 \times 224$ resolution and eight local views ($x_l$) at a lower resolution ($96 \times 96$). All of the views are processed by the student network, but the teacher network only processes the global views. The student and teacher networks both output $K$ dimensional feature vectors for each view, which are converted into probability distributions ($P_s$ and $P_t$) using a temperature scaled softmax function (using the parameters $\tau_s$ and $\tau_t$, respectively). The student network is trained to match the outputs of the teacher $P_t$ via stochastic gradient descent (SGD) with the cross-entropy loss: $J(\theta_s) = \sum_{x \in \{x_1^g, x_2^g\}} \sum_{\substack{x' \in \{x_1^g, x_2^g\} \cup x_l \\ x' \neq x}} P_t(x) \log(P_s(x'))$. The teacher network parameters are updated as an exponential moving average of the student network parameters.

Representations learned using a self-distillation loss are prone to collapse, meaning that the network converges to a trivial solution like predicting the same representation for every image. To avoid

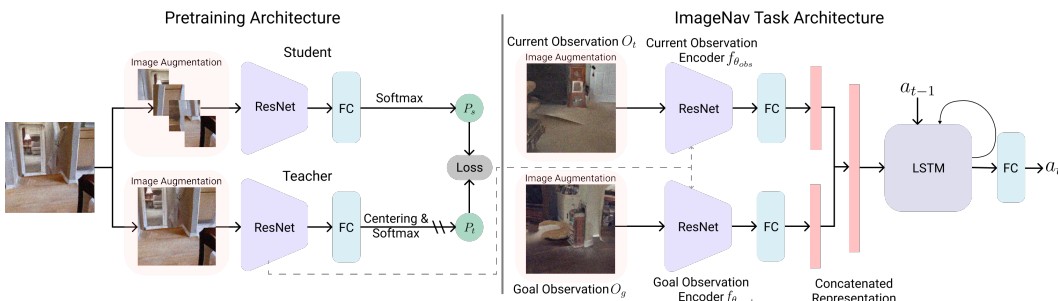

Figure 2: Overview of OVRL, consisting of two steps: 1) offline pretraining of the visual representations using large-scale pre-rendered images of indoor environments using DINO 2) downstream finetuning of the visuomotor representations on the IMAGENAV RL task in Habitat.

collapse, DINO centers and sharpens the teacher's output before the softmax operation. Specifically, the centering operation adds the term $c$ to the teacher's output, which is updated as follows: $c \leftarrow mc + (1 - m)\frac{1}{B}\sum_{i=0}^{B} g_{\theta_t}(x_i)$, where $m > 0$ is a momentum parameter and $B$ is the batch size. Sharpening is achieved by setting $\tau_t \ll 1$ for the teacher softmax normalization function.

We use the convolutional layers in a modified ResNet50 (He et al., 2016) architecture combined with a projection head for the student and teacher networks. Specifically, we modify the ResNet50 by 1) reducing the number of output channels at every layer by half (*i.e.* using 32 ResNet baseplanes instead of 64) and 2) using GroupNorm (Wu & He, 2018) instead of BatchNorm in our backbone, similar to DDPPO (Wijmans et al., 2020). The projection head is a 3-layer MLP (Multi-Layer Perceptron) with 2048 hidden units followed by $l_2$ norm and a weight normalized fully connected layer of K dim. We keep the BatchNorm in the projection head, however the head is discarded after training.

## 3.2 DOWNSTREAM LEARNING

The following paragraphs discuss the data augmentations used for policy learning and network architecture used for IMAGENAV and OBJECTNAV experiments.

**Data Augmentation:** Prior works (Laskin et al., 2020b; Yarats et al., 2020; 2021; Mezghani et al., 2021) have shown that using image augmentations during policy learning can help improve overall performance and leads to better generalization on the test set. We experimented with 4 different augmentations including color jitter, random rotation, random resized crop (Laskin et al., 2020b) and translate (Yarats et al., 2021). Similar to RAD (Laskin et al., 2020b), we use augmentations that are consistent over time, thus enabling the augmentation to retain temporal information. Also, in line with Mezghani et al. (2021), we use different augmentations when action sampling from the policy (for RL), and during the forward-backward pass (in both RL and IL).

**IMAGENAV Policy Learning:** Fig 2 shows our IMAGENAV architecture. At each timestep $t$, the policy $\pi(a_t|O_t, O_g)$ receives the current $O_t$ and goal observations $O_g$. These observations consist of RGB images, which are first passed through the data augmentation module and then featured using the observation $f_{\theta_{obs}}$ and goal $f_{\theta_{goal}}$ visual encoders. We initialize both encoders using ResNet50 weights from the pretraining stage. The feature vectors are then passed through a set of fully-connected layers, concatenated together and finally fed into a 2-layer, 512-dimensional LSTM network, along with the representation of the action from the previous timestep. The model is trained using DD-PPO (Wijmans et al., 2020).

**OBJECTNAV Policy Learning:** Fig 3 shows our OBJECTNAV architecture. We build upon Habitat-Web (Ramrakhya et al., 2022), which

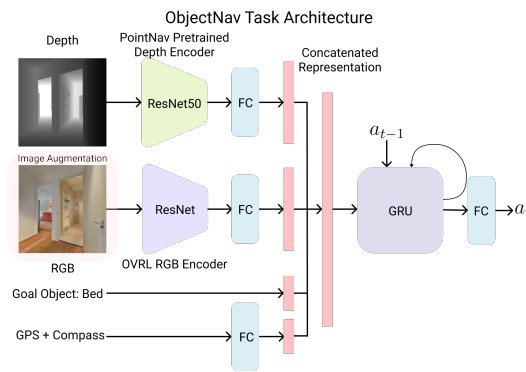

Figure 3: Our policy architecture for OBJECT-NAV task when using a RGBD camera and a GPS+Compass sensor. The RGB encoder is initialized with our pretrained weights.

| | Method | Pretraining Dataset | Test Split | Camera(s) | Test SPL (↑) | SR (↑) |
|---|---|---|---|---|---|---|
| 1) | Scratch | ✗ | A | 1 RGB | $9.3_{\pm1.1}\%$ | $17.9_{\pm2.0}\%$ |
| 2) | ZER (ResNet9) (Al-Halah et al., 2022) | ✗ | A | 1 RGB | $21.6\%$ | $29.2\%$ |
| 3) | ZER (ResNet50)* | ✗ | A | 1 RGB | $18.8_{\pm2.3}\%$ | $27.7_{\pm1.7}\%$ |
| 4) | CRL (Du et al., 2021) | MP3D | POINTNAV | 1 RGB | $3.2\%$ | $5.8\%$ |
| 5) | CRL* | Gibson | A | 1 RGB | $10.2_{\pm1.6}\%$ | $20.4_{\pm2.8}\%$ |
| 6) | OVRL (Ours) | OSD | A | 1 RGB | $26.9_{\pm0.9}\%$ | $41.3_{\pm1.0}\%$ |
| 7) | OVRL+ZER-Reward (Ours) | OSD | A | 1 RGB | $\mathbf{27.0_{\pm2.5}\%}$ | $\mathbf{54.2_{\pm1.4}\%}$ |
| 8) | Mem-Aug RL (Mezghani et al., 2021) | ✗ | A | 4 RGB | $56.0\%$ | $69.0\%$ |
| 9) | OVRL (Ours) | OSD | A | 4 RGB | $\mathbf{62.5_{\pm1.3}\%}$ | $\mathbf{79.8_{\pm0.7}\%}$ |
| 10) | NRNS (Hahn et al., 2021) | ✗ | B | 1 RGBD | $12.4\%$ | $24.0\%$ |
| 11) | OVRL (Ours) | OSD | B | 1 RGB | $\mathbf{28.4_{\pm1.7}\%}$ | $\mathbf{45.5_{\pm2.7}\%}$ |

Table 1: IMAGENAV performance of OVRL and baselines. (* Reimplementation)

uses imitation learning (IL) for training the
agent. The policy $\pi(a_t|O_t, G)$ receives the current observation $O_t$ at every timestep along with
the category ID of the goal object $G$. The current observation consists of a RGB + depth image and
the agent's location and orientation obtained from a GPS+Compass sensor. The RGB image is first
passed through the data augmentation module and then featurized using the RGB encoder $f_{\theta_{RGB}}$,
while the depth image is directly passed to the depth encoder $f_{\theta_{Depth}}$. We initialize the RGB en-
coder with our pretrained ResNet50 and finetune it during the task, while the depth encoder weights
are initialized with a ResNet50 pretrained on the POINTNAV task and kept frozen. These RGB and
depth features are then concatenated together with the GPS+Compass sensor and goal embeddings
and fed into a 2-layer 2048-dimensional GRU to predict a distribution over actions $a_{t+1}$. The model
is trained using a distributed version of behavior cloning (Ramrakhya et al., 2022).

## 4  EXPERIMENTS

In this section, we provide implementation details for our pretraining approach using DINO. Then,
we provide results of finetuning OVRL representation for the IMAGENAV and OBJECTNAV tasks
alongside comparisons with several baselines.

### 4.1  SELF-SUPERVISED PRETAINING

**Dataset.** We pretrain the ResNet encoder using Omnidata Starter Dataset (OSD) (Eftekhar et al.,
2021), consisting of approximately 14.5 million rendered images from 3D scenes: Replica (Straub
et al., 2019), Replica+GSO (ign), Hypersim (Roberts et al., 2021), Taskonomy (Zamir et al., 2018),
BlendedMVG (Yao et al., 2020) and Habitat-Matterport3D (HM3D) (Ramakrishnan et al., 2021b).
We exclude images from Gibson test scenes (which are part of Taskonomy) to not contaminate our
downstream experiments on IMAGENAV. We do not use the images from the CLEVR dataset (John-
son et al., 2017) due to their visual and domain dissimilarity from other scenes.

**Implementation Details.** We use the LARS optimizer (You et al., 2017) with weight decay $10^{-6}$
and batch size 8192, distributed over 64 GPUs. The learning rate starts at 0, is linearly increased
during the first 10 epochs to its base value 0.15, followed by a decay to the minimum value of
$5\times10^{-6}$ using a cosine schedule (Loshchilov & Hutter, 2017). The student and teacher temperatures
$\tau_s$ and $\tau_t$ are set to 0.1 and 0.04. The network is trained for a total of 100 epochs on OSD.

### 4.2  DOWNSTREAM LEARNING: IMAGENAV

We describe the implementation details related to the IMAGENAV task along with the details of the
training dataset in Appendix C.1 and introduce the baselines we adopt in this work in Appendix B.1.

In Table 1, we report IMAGENAV results averaged over 3 random seeds. In row 6, we show that
OVRL attains 41.3% SR and 26.9% SPL on split "A" (see Appendix C.1), while the scratch baseline
(row 1) only achieves 17.9% SR and 9.3% SPL. We emphasize that the only difference between

| Method | Pretraining Dataset | Camera(s) | Val SPL (↑) | Val SR (↑) | Test SPL (↑) | Test SR (↑) |
|---|---|---|---|---|---|---|
| 1) EmbCLIP (Khandelwal et al., 2022) | WIT | 1 RGB | - | - | **7.8**% | 18.1% |
| 2) Habitat-Web* | ✗ | 1 RGB | 4.6% | 17.4% | 4.9% | 15.1% |
| 3) OVRL (ours) | OSD | 1 RGB | 7.0% | 25.3% | 6.8% | **21.4**% |
| 4) DDPPO (Wijmans et al., 2020) | ✗ | 1 RGBD | 1.8% | 8.0% | 2.1% | 6.2% |
| 5) Habitat-Web (Ramrakhya et al., 2022) | ✗ | 1 RGBD | 6.1% | 22.7% | 6.1% | 17.0% |
| 6) Habitat-Web* | ✗ | 1 RGBD | 5.9% | 24.2% | 6.1% | 17.9% |
| 7) OVRL (ours) | OSD | 1 RGBD | **7.4**% | **28.6**% | **7.6**% | **23.2**% |

Table 2: OBJECTNAV results on MP3D VAL and TEST. (* Reimplementation)

these two methods is that OVRL is initialized with a pretrained encoder. This clearly shows that using pretrained encoders can make a big difference to the performance of an embodied agent.

We also find that OVRL outperforms all prior work. Specifically, OVRL outperforms ZER (Al-Halah et al., 2022) (row 2) by 12.1% in terms of success rate and 5.3% in SPL. In row 3, we find that reproducing ZER with a deeper CNN (ResNet50 vs. ResNet9) to match the visual encoder used in our approach leads to a small drop in ZER performance. However, in row 7, we find that using the ZER reward structure and goal view sampling with our pretraining approach (OVRL + ZER-Reward) further improves our agent's success rate by 12.9% and SPL by 1.1%. These results highlight that our pretraining approach can be used alongside other improvements to embodied agents to achieve additional performance gains. In row 4 and 5, we present results on CRL (Du et al., 2021), as reported in the paper and from our re-implementation. According to Du et al. (2021), CRL achieves 5.8% success rate and 3.2% SPL when pretrained on the MP3D dataset followed by finetuning and testing on the Gibson dataset using the POINTNAV episode splits. In comparison, our reimplementation achieves 20.4% success rate and 10.2% SPL on the test split "A". Comparing CRL with OVRL (row 6), we observe that CRL's alternative pretraining technique underperforms us by 20.9% in success rate and 16.7% in SPL.

In row 9, we extend OVRL to the multi-view setting (4 RGB) for direct comparison with Mem-Aug RL (Mezghani et al., 2021) (details in Appendix A.1). We find that OVRL outperforms Mem-Aug RL (Mezghani et al., 2021) by 10.8% in success rate and 6.5% in SPL despite not using an additional memory mechanism within the agent's architecture. Finally, in row 11, we evaluate on episodes used in Hahn et al. (2021) (split "B") and find that our approach outperforms NRNS (Hahn et al., 2021) by 21.5% in success and 16.0% in SPL even without having access to depth information or an egocentric pose estimate.

### 4.3 DOWNSTREAM LEARNING: OBJECTNAV

We describe the implementation details related to the OBJECTNAV task along with the details of the training dataset in Appendix C.2 and introduce the baselines we adopt in this work in Appendix B.2.

We compare our approach with existing OBJECTNAV methods in the RGB and RGBD observation settings and present our results for 1 seed in Table 2. On the MP3D TEST-STD split, in the RGB setting, we find that OVRL improves over Habitat-Web by 6.3% in success rate and 1.9% in SPL (row 2 *vs.* 3). OVRL also improves over EmbCLIP (Khandelwal et al., 2022) by 3.3% in success rate while performing 1.0% worse in SPL in the RGB setting (row 1 *vs.* 3). In the RGBD setting, OVRL's success rate and SPL on the test set improve to 23.2% and 7.6% respectively. Our implementation of Habitat-Web performs slightly better than the reported results in Ramrakhya et al. (2022) (row 5 *vs.* 6) since it uses a bigger network (ResNet18 vs ResNet50) and image augmentations. We observe that the gap between OVRL and Habitat-Web reduces to 5.3% and 1.5% for success rate and SPL respectively (rows 6 *vs.* 7); likely since the policy now relies lesser on the RGB image representations for navigation. OVRL also performs much better than DDPPO, achieving 17% higher success rate and 5.5% higher SPL (rows 4 *vs.* 7) in the RGBD seting.

## 5 ANALYSIS

In this section, we deconstruct our approach and systematically evaluate different design choices. We analyse the pre-training dataset choices, impact of data augmentations as well as the horizon of training in the following sections. The remainder of the analysis pertaining to model sizes and choice of pre-training algorithms is deferred to Appendix E. All analyses are conducted on the IMAGENAV task in the single RGB camera setting described in Section 4.2.

### 5.1 HOW MUCH DOES THE PRETRAINING IMAGE DATASET MATTER?

First, we evaluate the choice of the pretraining dataset and then study scaling laws for the best-performing dataset. We compare OSD with two alternative datasets: a) ImageNet-1k (Russakovsky et al., 2015), and b) Gibson-ShortestPath dataset, which consists of 12 million images collected (by us) from 307 Gibson scenes (Xia et al., 2018) using an oracle agent navigating the the POINTNAV training episodes via the shortest paths. Table 3 (rows 1-3) shows that representations learnt on the ImageNet-1k dataset achieve 20% success rate improving the Scratch baseline (from Table 1, row 1) by only 2.1%. Using the Gibson-ShortestPath dataset gives a significant boost to the performance, taking the success rate to 36.9%. This suggests that if the pretraining dataset is visually similar (though notice not identical) to the downstream task, the SSL objective is able to capture useful features. Also interestingly, we observe that the success rate of OVRL trained on OSD-100% (row 3) is higher than that Gibson-ShortestPath dataset by 4.4%. We attribute this improved performance to the diverse sampling scheme of the Omnidata pipeline.

Next, we study scaling laws for pretraining on OSD. We randomly subsample images from OSD and create smaller datasets containing only 1% (145K), 10% (1.45M) and 25% (3.625M) of the original, while ensuring that each successively smaller dataset is a subset of the larger ones. The results are summarized in Table 3 (rows 4-6). We notice that peak performance is achieved rather quickly, with essentially no difference between OSD-10%, 25%, and 100%. Notice that OSD contains images uniformly sampled from ∼2400 scenes; thus, we need somewhere between 60 images (1%) and 600 images (10%) per scene to learn useful representation for embodied tasks.

|  |  | Pretraining Dataset | Size | Test | |
|---|---|---|---|---|---|
|  |  |  |  | SPL ($\uparrow$) | SR ($\uparrow$) |
| DATASET | 1) | ImageNet-1k | 1.28M | $13.9_{\pm 2.8}\%$ | $20.0_{\pm 1.6}\%$ |
|  | 2) | Gibson-ShortestPath | 12.0M | $22.9_{\pm 1.2}\%$ | $36.9_{\pm 1.3}\%$ |
|  | 3) | OSD-100% | 14.5M | $\mathbf{26.9}_{\pm 0.9}\%$ | $\mathbf{41.3}_{\pm 1.0}\%$ |
| OSD SIZE | 4) | OSD-25% | 3.6M | $\mathbf{26.6}_{\pm 1.5}\%$ | $\mathbf{41.5}_{\pm 1.2}\%$ |
|  | 5) | OSD-10% | 1.45M | $26.6_{\pm 2.0}\%$ | $41.2_{\pm 0.8}\%$ |
|  | 6) | OSD-1% | 145K | $24.4_{\pm 1.3}\%$ | $37.6_{\pm 0.5}\%$ |
| OSD TYPE | 7) | OSD-Taskonomy | 4.5M | $27.0_{\pm 1.8}\%$ | $40.7_{\pm 1.6}\%$ |
|  | 8) | OSD-HM3D | 9.5M | $26.6_{\pm 0.1}\%$ | $41.6_{\pm 0.1}\%$ |
|  | 9) | OSD-Taskonomy+HM3D | 14.0M | $\mathbf{27.4}_{\pm 1.1}\%$ | $\mathbf{42.4}_{\pm 0.3}\%$ |

Table 3: OVRL's performance on IMAGENAV with different datasets, dataset sizes and types.

Finally, since OSD contains images from multiple 3D scene datasets (Taskonomy/Gibson, HM3D, Replica+GSO, BlenderMVG), we study the contribution of its constituents in Table 3 (rows 7-9). We find that using only HM3D or Taskonomy subset for pretraining gives similar performance as using the full dataset, while using the combination (Taskonomy + HM3D) leads to a slight increase in the performance. We hypothesize this gain in performance between OSD-100% (row 3) and OSD-Taskonomy+HM3D (row 9) is due to the removal of the images from the Replica+GSO (Eftekhar et al., 2021; ign) and BlendedMVG (Yao et al., 2020) datasets, which contain a large number of images focused on household objects and sculptures, therefore deteriorating the quality of the learnt representation for IMAGENAV.

### 5.2 HOW DO AUGMENTATIONS AND FINETUNING IMPACT PERFORMANCE?

In this section, we disentangle the contributions of finetuning the vision encoder *from* image augmentations (during finetuning stage). We do this by freezing the visual encoder during IMAGENAV training and refer to this as OVRL-Frozen. We then analyse performance of models re-trained and tested without augmentations. Table 4 shows the results (notice that Scratch and OVRL results in the presence of augmentations are identical to those in Table 1). First, we notice that finetuning is an

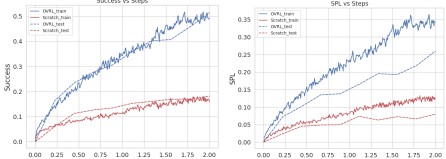

| | Test | |
|---|---|---|
| Method | SPL (↑) | SR (↑) |
| Scratch-HM3D | 8.0% | 19.3% |
| OVRL-HM3D (Ours) | 26.0% | 51.4% |

Figure 4: IMAGENAV Success/SPL vs. Steps for Scratch (red) and OVRL (blue) on HM3D training (solid) and Gibson testing (dashed).

Table 5: Comparison of OVRL against Scratch in the IMAGENAV task when trained on HM3D scenes and tested on Gibson test scenes

unconditionally good idea (with and without augmentation), with *significant* gains in the presence of augmentation – +11.2% success rate and +9.9% SPL (row 2 *vs*. 3, col. "Augmentations"). This result is not surprising, since finetuning allows the visual representations to adapt to a novel image distribution and capture task-specific details.

On the other hand, augmentations seem to be conditionally good. Adding augmentations during finetuning while keeping the vision encoder frozen hurts – the success rate and SPL drop by 2.9% and 2.0% respectively (row 2). We speculate that augmentations are harmful for OVRL-Frozen since the network is incapable of learning the useful invariances from the image augmentations. However, if the vision encoder is finetuned, using augmentations improve success rate and SPL by 6.0% and 3.8 (row 3)%. Finally, we see that the pretraining step is leading to 12-16% success rate improvement over Scratch (row 1 *vs*. 2). This shows that both finetuning and image augmentations are important components for achieving the best performance of OVRL, but augmentations should not be used alone.

### 5.3 IS OVRL EFFECTIVE WHEN THE AGENT IS TRAINED FOR MUCH LONGER?

Performance gains of pretrained models are known to diminish (or entirely disappear) when finetuned with long schedules (He et al., 2019). So, are OVRL representations also just useful for short finetuning schedules? To answer this question, we train an OVRL agent on the IMAGENAV task using the 800 training scenes from the HM3D dataset (Ramakrishnan et al., 2021b) and compare against an agent trained from scratch. We use the 8 million POINTNAV training episodes generated by Ramakrishnan et al. (2021b) to train for 2 billion simulation steps and then present the test performance on the Gibson test split "A" (Mezghani et al., 2021) in Table 5 for 1 seed. We find a high correlation between the training and testing success rates of the two methods in Fig 4, indicating positive transfer from the HM3D to Gibson dataset. We further observe that OVRL-HM3D even surpasses the performance of OVRL trained on the Gibson scenes (Table 1 row 6). Finally, in Fig 4, we observe that the benefits of OVRL pretraining over scratch are not only sustained, but continue to *increase* over the course of this long training schedule, which suggests that a significant rethinking might be needed in what is consider to be the 'standard' training schedule for these tasks.

## 6 CONCLUSIONS

We presented OVRL, a simple two-stage technique for using self-supervised learning techniques in EAI tasks. We show that OVRL learns generalizable representations and present results on the IMAGENAV and OBJECTNAV tasks in Habitat. OVRL achieves a success rate of 54% on IMAGENAV in Gibson (+25% absolute improvement in SOTA) in the single-RGB setting and 23% on OBJECTNAV in MP3D (+5.1% absolute improvement in SOTA) in the RGB/RGBD-only setting. Finally, we conduct extensive ablation studies to understand the usefulness of the different components in our approach.

| | Method | Augmentations | | No Augmentations | |
|---|---|---|---|---|---|
| | | SPL (↑) | SR (↑) | SPL (↑) | SR (↑) |
| 1) | Scratch | $9.3_{\pm1.1}$% | $17.9_{\pm2.0}$% | $9.3_{\pm0.7}$% | $16.9_{\pm1.4}$% |
| 2) | OVRL-Frozen | $17.0_{\pm0.7}$% | $30.1_{\pm1.6}$% | $19.0_{\pm0.1}$% | $33.0_{\pm0.4}$% |
| 3) | OVRL | $\mathbf{26.9_{\pm0.9}}$**%** | $\mathbf{41.3_{\pm1.0}}$**%** | $23.1_{\pm0.4}$% | $35.3_{\pm0.4}$% |

Table 4: IMAGENAV performance with/without augmentations and finetuning.

## ACKNOWLEDGEMENTS

The Georgia Tech effort was supported in part by NSF, ONR YIP, and ARO PECASE. The views and conclusions contained herein are those of the authors and should not be interpreted as necessarily representing the official policies or endorsements, either expressed or implied, of the U.S. Government, or any sponsor.

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

# Appendix

## A APPROACH

### A.1 4 CAMERAS EXPERIMENT: NETWORK ARCHITECTURE

To compare against Mem-Aug Nav (Mezghani et al., 2021), we take the architecture proposed in Section 3 from main paper and extend it to work in the multi-view (4 RGB Cameras) setting. At each timestep $t$, the policy $\pi(a_t|O_t, O_g)$ receives the current observation $O_t$ and goal observation $O_g$. Both $O_t$ and $O_g$ tuples consist of 4 RGB images from 4 cameras facing facing 'front', 'back', 'left', and 'right' directions (Mezghani et al., 2021). These observations are first passed through the data augmentation module and then featurized using the observation $f_{\theta_{obs}}$ and goal $f_{\theta_{goal}}$ visual encoders. We initialize both encoders using ResNet50 weights from the pretraining stage. The current observation feature vectors are then passed through a fully-connected (FC) layer, concatenated together and passed through another FC layer. For the goal representation, the feature vectors are passed through an FC layer and the outputs are added together similar to Mezghani et al. (2021) to keep the representation rotation-invariant. Finally, the current and goal observation representations are concatenated together and fed into a 2-layer, 512-dimensional LSTM network, along with the representation of the action from the previous timestep.

## B BASELINES

### B.1 IMAGENAV

On the IMAGENAV task, we compare OVRL against the following baselines:

– **Scratch**: A baseline identical in structure and training to OVRL except trained from scratch without any pretraining. Comparing OVRL to Scratch establishes the value of pretraining.
– **Curious Representation Learning (CRL)**: CRL (Du et al., 2021) is a two-stage approach that pretrains a visual encoder under the SimCLR (Chen et al., 2020) objective using images collected online while training with an exploration policy that is rewarded to maximize the SSL loss. The encoder is then frozen and used for downstream IMAGENAV task. Unfortunately, the CRL manuscript (Du et al., 2021) reported results with 10M frames of training, which we find to be deficient for 2 reasons – in our experience, training does not converge till 500M frames and results at 10M are highly sensitive to initialization. Thus, we reimplemented and engineered CRL to work on multiple GPUs and scaled up CRL pretraining and training phases from 10M steps to 500M in Gibson scenes. Consequently, the results reported here are significantly stronger than those presented in Du et al. (2021). Comparing OVRL to CRL establishes the value of pretraining on an IID-sampled (vs agent gathered) dataset of images, while disentangling the effects of scaling.
– **Zero-Experience Required (ZER)**: ZER (Al-Halah et al., 2022) extends the Scratch baseline by introducing a new reward structure that encourages matching the orientation of the goal image. Additionally, ZER randomly samples goal image orientations during training to increase the number of novel episodes.
– **Mem-Aug RL** (Mezghani et al., 2021): Memory-augmented RL proposes an attention based model that leverages episodic memory for navigation with panoramic images.
– **NRNS** (Hahn et al., 2021): NRNS uses passive videos to learn a geodesic distance estimator and a target prediction model to create a topological map and then navigates an agent using a combination of global and local policy. It requires both an RGBD image and egocentric pose information to navigate.

### B.2 OBJECTNAV

On OBJECTNAV we compare our method against the following baselines:

– DDPPO: A standard RL baseline that uses a single ResNet18 encoder with 32 baseplanes for RGB+Depth observations trained with DD-PPO (Wijmans et al., 2020).
– EmbCLIP: EmbCLIP (Khandelwal et al., 2022) investigates the effectiveness of CLIP's (Radford et al., 2021) visual representation for EAI tasks. The policy encodes the RGB observations using CLIP's frozen ResNet50, which are combined with a goal embedding and passed through a two-

layer CNN to generate a goal-conditioned visual embedding. These embeddings are finally passed through a recurrent model and the policy is trained using DD-PPO (Wijmans et al., 2020).

– Habitat-Web (Ramrakhya et al., 2022): We reimplement the agents from Habitat-Web to use identical architecture and image augmentations as OVRL, except the RGB encoder is trained from scratch. This comparison establishes the value of pretraining.

## C  IMPLEMENTATION DETAILS

### C.1  IMAGENAV

**Datasets.** We conduct experiments using the Habitat simulator (Savva et al., 2019; Szot et al., 2021) and the Gibson dataset (Xia et al., 2018), on the standard set of 72 training and 14 testing scenes. Unlike POINTNAV and OBJECTNAV, IMAGENAV does not yet have an authoritative benchmark and prior works often report results in settings with subtle differences, making results incommensurable. We report results under multiple settings and datasets to allow direct comparisons to a variety of prior work. Specifically, we report results on two sets of evaluation episodes – (1) split "A" contains 4,200 episodes (300 / testing scene) and was generated by Mezghani et al. (2021), (2) split "B" contains 3,000 episodes (approximately 200 / testing scene) and was generated by Hahn et al. (2021).

**Task Details.** Our IMAGENAV agents are equipped with only RGB cameras ($128 \times 128$-resolution), with no access to depth or GPS+Compass sensors (though we do compare to and outperform prior work that uses depth cameras). We consider two variants – (1) one front-facing camera (1 RGB) and (2) four cameras (4 RGB) facing 'front', 'back', 'left', and 'right' directions (as in Mezghani et al. (2021)). The agent's action space consists of four actions: MOVE_FORWARD ($0.25m$), TURN_LEFT ($30°$), TURN_RIGHT ($30°$) and STOP. An episode is terminated if the agent calls the STOP action or after the agent has taken 1000 steps in the environment. If the STOP action is called within $1m$ of the goal location, the episode is a success. We report the success rate (SR) and Success weighted by Path Length (SPL) (Anderson et al., 2018), which is a measure of path efficiency. Agents are trained for 500M steps (25k updates) using 32 GPUs with 10 environments each. Every worker takes (up to) 64 steps, followed by 2 PPO epochs with 2 mini-batches using a learning rate of $2.5 \times 10^{-4}$.

**Data Augmentation.** For IMAGENAV, we first found the best set of augmentations by conducting sweeps over individual augmentation parameters and then trying the best sequence of two data augmentations. We found applying color jitter with a value of 0.3 for brightness, contrast, saturation and hue levels followed by translation (Yarats et al., 2021) with a pad of 4 pixels on all the images gave us the best results. We also found data augmentations useful during testing and thus, we report all our results with test-time augmentations.

### C.2  OBJECTNAV

**Dataset.** We conduct OBJECTNAV experiments in the Habitat simulator (Savva et al., 2019; Szot et al., 2021) using the Matterport3D (MP3D) scene dataset (Chang et al., 2017), with the standard set of 61 training, 11 validation and 18 testing scenes. The results are reported on the MP3D VAL and TEST-STD split from Habitat Challenge OBJECTNAV benchmark.

**Task Details.** In the OBJECTNAV task, an agent is tasked with navigating to an instance of a specified object category (*e.g.* 'sofa') in a unseen environment. The agent does not have access to a map of the environment and must navigate with RGBD camera and GPS+Compass sensor which provides location and orientation information relative to the start of the episode. Visual observations from the simulator are first downsampled by 0.5, from $480 \times 640$ to $240 \times 320$, and then resized and center cropped to $256 \times 256$ while preserving the aspect ratio. Additionally, the agent also receives the goal object category ID as input. The agent's action space consists of MOVE_FORWARD ($0.25m$), TURN_LEFT ($30°$), TURN_RIGHT ($30°$), LOOK_UP ($30°$), LOOK_DOWN ($30°$), and STOP actions. For an episode to be considered success the agent has to stop within $1m$ euclidean distance of the goal object within 500 steps and be able to turn to view the object from the end position. We train all agents for ∼475M steps. We evaluate checkpoints at every ∼30M steps for last 240M steps of training and report metrics for the checkpoints with the highest success on the validation split.

**Imitation Learning.** We use the dataset of OBJECTNAV human demonstrations collected by Habitat-Web (Ramrakhya et al., 2022). For each of the 56 Matterport3D train scenes and each goal

category, it provides approximately $\sim 42$ demos from randomly-chosen start locations. We perform behavior cloning on $40k$ human demos, which amounts to $\sim 13.7$M frames of experience.

**Data Augmentation.** Similar to IMAGENAV, in OBJECTNAV we use a combination of color jitter followed by translation. We did a grid search over the hyperparams and found that color jitter values of 0.4 for brightness, contrast, saturation and hue levels followed by a pad of 16 pixels gave the best results.

## D ADDITIONAL EXPERIMENTS

In this section, we present results on additional experiments we conducted on the IMAGENAV and OBJECTNAV task along with indoor scene classification experiments.

### D.1 DOWNSTREAM LEARNING: INDOOR SCENE CLASSIFICATION

In this section, we study the representations learned with OVRL pretraining and after downstream finetuning to gain insights into our two-stage learning framework. Specifically, we evaluate our pretrained models on an indoor scene classification task.

**Implementation Details.** For this experiment, we use a subset of the Places365-Standard dataset (Zhou et al., 2017) containing images from 55 categories corresponding to indoor room scenes (details in Section G). To evaluate the representation, we use the pretrained visual encoders (ResNet50 with 32 baseplanes) from different methods, freeze their weights and learn a linear classifier on the final average pooled features obtained from the encoder. We train all baselines for 60 epochs using Adam optimizer with a learning rate of $1e - 3$.

**Baselines.** We compare OVRL against visual encoders obtained from the IMAGENAV from Scratch baseline and CRL (Du et al., 2021) presented in Section 4.2 of the main paper. We also compare with two encoders pretrained on the ImageNet dataset (Russakovsky et al., 2015). The first is pretrained with DINO, the same SSL algorithm used in OVRL pretraining. The second is pretrained with supervised learning using the ImageNet labels.

**Results.** We report the top-1 and top-5 accuracy on the Places dataset in Table 6. We find that OVRL pretraining (OVRL-Frozen – row 5) outperforms alternative pretraining methods or pretraining on other datasets. Specifically, OVRL-Frozen achieves $82.9\%$ top-5 accuracy and $50.9\%$ top-1 accuracy, which is $18.9\%$ better in terms of top-1 accuracy compared to the representations learnt by the Scratch baseline on IMAGENAV task (row 5 *vs*. 1). Our approach also performs $8.8\%$ better in top-1 accuracy when compared to our implementation of the pretraining approach proposed in CRL (Du et al., 2021). Next, we show that using DINO (Caron et al., 2021) (the same SSL method as OVRL) to pretrain on ImageNet (Russakovsky et al., 2015) dataset doesn't lead to representations that transfer as well as us to

|   | Method | Val Accuracy | |
|---|--------|-------|-------|
|   |        | Top 1 | Top 5 |
| 1) | IMAGENAV from Scratch | 32.0% | 62.7% |
| 2) | CRL(Du et al., 2021) | 21.2% | 48.8% |
| 3) | CRL* | 42.1% | 74.0% |
| 4) | ImageNet-DINO | 48.3% | 80.4% |
| 5) | OVRL-Frozen | **50.9%** | **82.9%** |
| 6) | OVRL | 42.9% | 76.0% |
| 7) | ImageNet-SL | 54.7% | 85.7% |

Table 6: OVRL's visual encoder compared against baselines on the classification task using the Places dataset. (* Reimplementation)

this task. The ImageNet-DINO baseline performs $2.6\%$ worse compared to OVRL-Frozen (row 4 *vs*. 5). Interestingly, we find that finetuning on the IMAGENAV task reduces scene classification performance by $8\%$ (rows 5 *vs*. 6). This suggests that some of the visual features learned through SSL that are useful for high-level scene classification, might not be helpful for IMAGENAV. Finally, we compare OVRL with representations learnt using supervised learning on ImageNet, and find that ImageNet-SL is $3.8\%$ better in top-1 accuracy (row 5 *vs*. 7).

### D.2 DOWNSTREAM LEARNING: IMAGENAV

**ImageNet Pretrained Models** In Section 5.1 of the main paper, we compared the performance of models pretrained with DINO using ImageNet-1k dataset, Gibson-ShortestPath dataset and OSD.

| | | Test | |
|---|---|---|---|
| Method | Augs. | SPL (↑) | SR (↑) |
| 1) Scratch | ✓ | 9.3% | 17.9% |
| 2) ImageNet-Finetuned | ✓ | 13.9% | 20.0% |
| 3) ImageNet-Frozen | ✓ | 13.3% | 23.7% |
| 4) ImageNet-Frozen | ✗ | **14.9%** | **26.1%** |

Table 7: Performance of ImageNet pretrained visual encoders without finetuning and augmentations on IMAGENAV.

We find it surprising that ImageNet pretraining leads to only a slight improvement in performance (+2.1% success and +4.6%) over Scratch baseline and therefore investigate this issue further. We start by freezing the pretrained visual encoder and see a 3.7% improvement in the success rate (row 1 *vs.* 2, Table 7) and 0.6% deterioration in the SPL. We follow this by removing the image augmentations from the frozen baseline and find an overall improvement of 6.1% in success rate and 1% in the SPL over Scratch (row 1 *vs.* 3). Overall, these results demonstrate that ImageNet representations are not easily adapted to embodied tasks such as IMAGENAV using simple finetuning and data augmentation techniques. In contrast, the performance of our approach (row 6, Table. 1 in the main paper), which is pretrained on OSD, improves with downstream finetuning and can take advantage of data augmentation.

# E   ADDITIONAL ANALYSIS

## E.1   HOW DOES THE CHOICE OF SSL ALGORITHM AFFECT PERFORMANCE?

While our approach is agnostic to the choice of SSL algorithm, we conduct an ablation to show that our choice of SSL algorithm, DINO, is superior to other existing approaches. We compare against MOCO-v2 and SimCLR, two well-known contrastive learning techniques. On the ImageNet classification task, DINO is known to outperform both these approaches in terms of the linear and $k$-NN classification accuracy. In Table 8, we find that DINO surpasses both algorithms in terms of success and SPL on the IMAGENAV tasks as well. In the future, OVRL can be coupled with improved SSL algorithms.

| SSL Algo. | Test | |
|---|---|---|
| | SPL (↑) | SR (↑) |
| 1) MOCOv2 | $23.7_{\pm1.6}\%$ | $35.1_{\pm0.7}\%$ |
| 2) SimCLR | $22.8_{\pm0.8}\%$ | $38.1_{\pm1.2}\%$ |
| 3) DINO | $\mathbf{26.9}_{\pm0.9}\%$ | $\mathbf{41.3}_{\pm1.0}\%$ |

Table 8: Comparison of different SSL algorithms on IMAGENAV.

| Vision Encoder | Test | |
|---|---|---|
| | SPL (↑) | SR (↑) |
| 1) ResNet18-32B | $21.9_{\pm2.3}\%$ | $33.7_{\pm0.9}\%$ |
| 2) ResNet50-32B | $26.9_{\pm0.9}\%$ | $41.3_{\pm1.0}\%$ |
| 3) ResNet50-64B | $\mathbf{27.5}_{\pm0.8}\%$ | $\mathbf{41.8}_{\pm0.9}\%$ |
| 4) ResNet101-32B | $26.4_{\pm1.4}\%$ | $40.8_{\pm0.1}\%$ |

Table 9: Ablation of the model size on IMAGENAV task

## E.2   DOES MODEL SIZE MATTER?

We study the choice of our vision model (ResNet50-32B) by comparing it against a ResNet18 and a ResNet101 network with 32 baseplanes and a ResNet50 with 64 baseplanes. We see in Table 9 that using a ResNet18 leads to more than 7% decline in success rate and 5.0% decline in SPL (row 1 vs. 2) as the model fails to consolidate all the knowledge in a smaller network. On the other hand, increasing the number of layers or baseplanes leads to increased compute requirements, with only a slight improvement in performance for ResNet50-64B (row 3).

## E.3   DOES THE SCRATCH BASELINE CATCH UP WITH OVRL ON GIBSON SCENES?

In Section 5.5 in the main paper, we studied OVRL's properties while training on the HM3D dataset for a long time horizon (2 billion (B) simulation steps). Here, we conduct a similar study for the Gibson training scenes (Mezghani et al., 2021) in the IMAGENAV task, by training OVRL and Scratch

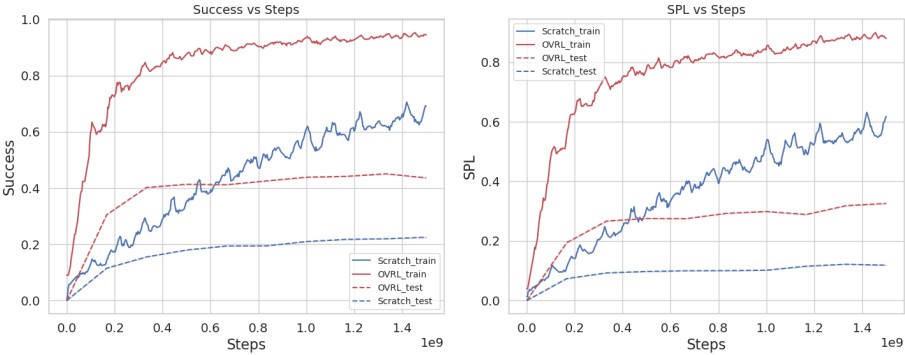

Figure 5: IMAGENAV Success/SPL vs. Steps for Scratch (red) and OVRL (blue) on Gibson training (solid) and Gibson testing (dashed) scenes for 1.5 billion steps.

agent for 1.5B steps. Gibson being a much smaller dataset compared to HM3D (72 vs 800 training scenes), we expect to see a faster convergence of the two methods on Gibson. In Fig. 5, we observe that the OVRL starts to converge after 500 million (M) steps on the training set, finally reaching a success rate of 95% and SPL of 89% at the end of training. Meanwhile, Scratch baseline continues to improve after 500M steps, with both the success rate and SPL nearly doubling in the next 1B steps of training to final values of 70% and 60% respectively. However, on the test set we see much weaker gains, with Scratch only improving by 5% in success rate and 4.2% in SPL, reaching final values of 22.9% and 13.5% respectively. In comparison, OVRL's success rate of 41.3% and SPL of 26.9% at 500M steps (row 6, Table 1 in the main paper), is still approximately double the current performance of Scratch. Overall, this experiment demonstrates that Scratch is unable to match OVRL's performance even with significantly longer training horizon and supports the hypothesis about importance of offline visual representation.

### E.4 HOW DO SSL REPRESENTATIONS COMPARE AGAINST THE REPRESENTATIONS LEARNT WITH RL ON A GIVEN TASK?

To answer this question, we take the visual encoder learnt by the Scratch agent on IMAGENAV and use it as the pretrained visual encoder for a downstream task. We choose the same IMAGENAV task for this downstream experiment which eliminates the possibility of encountering any kind of distribution shift in the dataset. We try both finetuning and freezing the RL pretrained visual encoder and call these methods RL-Pretrained-Finetuned and RL-Pretrained-Frozen respectively. Looking at the results in Fig 6, we observe a significant improvement in the success rate and SPL of both the methods over the Scratch baseline on the training episodes. We also see that the gap between the success rate of OVRL and RL-Pretrained baselines reduces to only 8%. However, as we look at the test performance, we observe a significant generalization gap in the performance of RL-Pretrained methods. Finally, comparing RL-Pretrained methods and OVRL (Table 11) on the test set, we see a difference of approximately 14% in success rate and 11% in SPL. Overall, this result demonstrates that pretraining visual encoders using self-supervised learning is beneficial for generalization in embodied AI tasks.

| Method | Test | |
| --- | --- | --- |
| | SPL (↑) | SR (↑) |
| Scratch | 13.5% | 22.9% |
| OVRL (Ours) | 33.4% | 45.5% |

Table 10: Comparison of OVRL against Scratch in the IMAGENAV task when trained and tested on Gibson scenes for 1.5 billion steps

| Method | Test | |
| --- | --- | --- |
| | SPL (↑) | SR (↑) |
| Scratch. | 9.3% | 17.9% |
| RL-Pretrained-Finetuned | 15.1% | 27.3% |
| RL-Pretrained-Frozen | 16.1% | 27.2% |
| OVRL (Ours) | **26.9%** | **41.3%** |

Table 11: Comparison of OVRL and Scratch against RL-Pretrained baselines in the IMAGENAV task

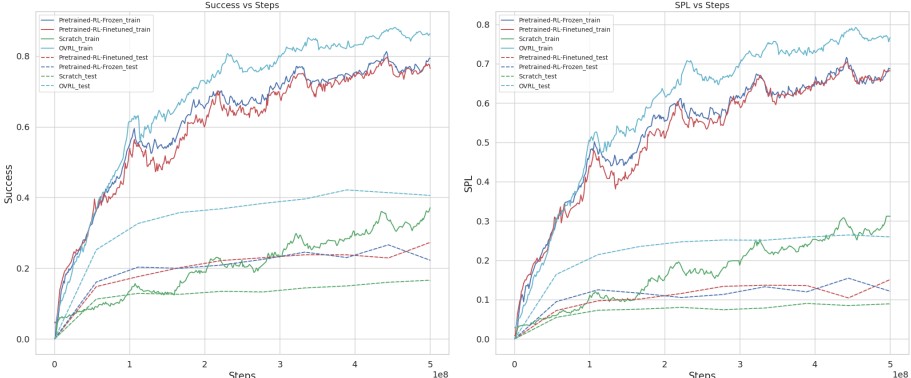

Figure 6: IMAGENAV Success/SPL vs. Steps for Scratch (green), OVRL (cyan) and the RL-Pretrained (red, blue) methods on the Gibson train (solid) and test (dashed) scenes.

## F    COMPUTE RESOURCES

We use the 32GB Tesla V100 GPU for all our experiments. Pre-training our vision encoder on the 12 million images from the Omnidata Starter Dataset takes a total of 65 hours on 64 GPUs – *i.e.* 173 GPU days of training. A single run of OVRL using the default settings on the ImageNav task takes 40 hours to train parallelly on 32 GPUs – equivalent to 53 GPU days. The ImageNav experiments in Section 4.2 take approximately 960 GPU days to train in total. In ObjectNav, we train for 2 days in our RGB only experiments and 3 days in the RGBD experiments, taking a total of 640 GPU days. Finally, the experiments across the analysis section total over 3500 GPU days of training time.

## G    PLACES DATASET

For the scene classification experiments in Section D.1, we use 55 indoor scene classes from the Places365 dataset (Zhou et al., 2017) adapted from Du et al. (2021). Specifically, we use the following classes: airport_terminal, apartment_building-outdoor, art_gallery, art_studio, attic, auditorium, ballroom, banquet_hall, bar, basement, beauty_salon, bedroom, bookstore, cafeteria, classroom, closet, cockpit, coffee_shop, conference_center, conference_room, corridor, dining_room, dorm_room, engine_room, fire_escape, home_office, hospital_room, hotel-outdoor, hotel_room, inn-outdoor, kitchen, living_room, lobby, locker_room, mansion, martial_arts_gym, motel, museum-indoor, music_studio, nursery, office, office_building, patio, railroad_track, residential_neighborhood, restaurant, restaurant_kitchen, restaurant_patio, shed, shopfront, shower, stage-indoor, staircase, swimming_pool-outdoor, waiting_room.

