# OpenReview forum: "Offline Visual Representation Learning for Embodied Navigation"
_ICLR.cc/2023/Workshop/RRL — RRL 2023 Poster_

### Official Review · Reviewer_vy8T · 2023-03-01
**Interesting paper on pretraining representations for navigation**

**Rating:** 4
**Confidence:** 3

**Review:**

This paper presents a 2-stage procedure to a) learn representations from offline data b) fine-tuning on downstream tasks. The method builds on DINO, a self-supervised algorithm relying on distillation.
The paper demonstrates the benefit of the approach on on 3 different 3D datasets and 2 policy learning algorithms (RL, IL). The paper is clearly written and contains many ablations (benefit of data augmentation / effect of dataset size / long fine-tuning schedules).
Overall, this paper would be a nice contribution to the workshop.

---

### Official Review · Reviewer_a5Ae · 2023-03-01
**A valid contribution to the RL/IL community for how to do visual encoder pretraining better for visual navigation**

**Rating:** 3
**Confidence:** 4

**Review:**

This paper studies the pretraining strategies regarding the visual perception capabilities of RL/IL agents for visual navigation tasks. It proposes to leverage a very large-scale dataset of rendered images from 3D scenes with recently proposed self-supervised learning techniques (i.e., DINO). The authors did a good job examining the key components of both their pretraining approach and the follow-up online fine-tuning process, with large improvements brought by their method. The specific dataset that is visually similar to the downstream tasks seems to be one of the keys besides good data augmentation strategies. However, while this method is proposed for visual navigation, how to do pertaining better for more complex Embodied AI tasks beyond navigation still remains an open question, as pointed out by another recent paper [1].


[1] On Pre-Training for Visuo-Motor Control: Revisiting a Learning-from-Scratch Baseline